# Modified Clavien–Dindo Classification and Outcome Prediction in Free Flap Reconstruction among Patients with Head and Neck Cancer

**DOI:** 10.3390/jcm9113770

**Published:** 2020-11-22

**Authors:** Wei-Ling Jan, Hung-Chi Chen, Chang-Cheng Chang, Hsin-Han Chen, Pin-Keng Shih, Tsung-Chun Huang

**Affiliations:** 1Department of Surgery, China Medical University Hospital, Taichung 404332, Taiwan; willyjan58@gmail.com; 2International Center, China Medical University Hospital, Taichung 404332, Taiwan; d19722@mail.cmuh.org.tw; 3Division of Plastic and Reconstructive Surgery, Department of Surgery, China Medical University Hospital, Taichung 404332, Taiwan; d13749@mail.cmuh.org.tw (H.-H.C.); d20827@mail.cmuh.org.tw (P.-K.S.); d21920@mail.cmuh.org.tw (T.-C.H.); 4School of Medicine, College of Medicine, China Medical University, Taichung 404333, Taiwan; 5Institute of Imaging and Biomedical Photonics, National Chiao Tung University, Tainan 71150, Taiwan

**Keywords:** Clavien–Dindo classification, head and neck cancer, free flap reconstruction

## Abstract

Because of limitations caused by unique complications of free flap reconstruction, the Clavien–Dindo classification was modified to include grade “IIIc” for “partial or total free flap failure.” From 2013 to 2018, 116 patients who had undergone free flap reconstruction for head and neck cancer with grade III or higher postoperative complications were grouped using the “Modified” Clavien–Dindo classification. Alcoholism displayed significant predictive effects between grade IIIb and IIIc (72.7% vs. 50%, *p* = 0.028). Significant differences were observed between grade IIIb and IIIc in the duration of hospital stay (23.1 ± 10.1 vs. 28.6 ± 11.9 days, *p* = 0.015), duration of intensive care unit stay (6.0 ± 3.4 vs. 8.7 ± 4.3 days, *p* = 0.001), reoperation times during the current hospitalization (1.4 ± 0.8 vs. 2.0 ± 1.0 times, *p* < 0.001), and wound infection rate (29.9% vs. 62.5%, *p* = 0.002). The severity levels were significantly positively correlated with reoperation times during the current hospitalization (*p* < 0.001), ICU stay (*p* = 0.001), and hospital stay (*p* < 0.001). The modified Clavien–Dindo classification with grade IIIc describes the perioperative complications of head and neck free flap reconstruction to predict clinical outcomes based on severity.

## 1. Introduction

A standardized grading system of surgical complications is necessary for improving the quality of perioperative patient care, decision making, and the consistency of interpretation between professionals. The Clavien–Dindo classification is widely used in general and urologic surgery because of its validity and reliability [1,2,3,4,5,6]. It was first described by Clavien and Dindo in 2004 [1]. Grade I of the Clavien–Dindo classification was defined as any deviation from the normal postoperative course without the need for pharmacological treatment or surgical, endoscopic, or radiological interventions. Acceptable regimens include antiemetics, analgesics, antipyretics, diuretics, and bedside wound care. Grade II was defined as complications requiring pharmacological treatment with drugs other than those permitted for grade I complication. Grade III of the Clavien–Dindo classification was originally defined as surgical complications requiring surgical, endoscopic, or radiological intervention, and was further classified into grade IIIa and IIIb depending on whether the intervention was performed under general anesthesia. Grade IV was defined as a life-threatening complication that requires management in a high dependency and was further classified into grade IVa and IVb in terms of single or multi-organ dysfunction. Finally, a complication resulting in death was graded as V. However, this classification is insufficient for use in several specific surgical fields, including orthopedic surgery, thoracic surgery, and pediatric urology. Therefore, a growing number of modified systems are being investigated and published in various surgical fields [4,7,8,9].

Composite resection followed by free flap reconstruction has become the standard treatment for patients with head and neck cancer. However, the specific complications of head and neck free flap reconstruction, such as hematoma evacuation, fistula formation, and flap failure, lack clarification, which limits the use of conventional classification [3,4,10]. For example, grade IIIb refers to complications requiring interventions under general anesthesia. Consequently, relatively minor complications, such as hematoma evacuation, and severe complications, such as partial or complete flap loss, requiring additional salvage procedures, would be assigned the same grade. However, there are marked differences between the treatment and eventual outcomes of minor and severe complications [4].

An ideal and broadly accepted grading system for the postoperative complications of head and neck free flap surgery has not yet been developed. Therefore, we modified the conventional classification by adding the grade “IIIc” for “’partial or total free flap failure after intervention’ needing further surgery in general anesthesia.” Several surrogate factors, including the duration of hospital stay, duration of intensive care unit (ICU) stay, and hospital costs, were employed to assess the effect of postoperative complications on surgical outcomes [2,10,11,12,13]. Herein, we investigated multiple comorbidities for outcome prediction and the correlation between the modified classification and the severity level.

## 2. Material and Methods

This retrospective cohort study was approved by the Institutional Review Board of China Medical University Hospital (No. CMUH108-REC1-021). From November 2013 to December 2018, data were collected from 120 patients with head and neck cancer who had undergone an unplanned second operation after free flap reconstruction during hospitalization at China Medical University Hospital. After a complete review of the electronic medical charts of each patient, four patients were excluded because regional flaps were used rather than free flaps in these patients. Therefore, 116 patients with head and neck cancer, who were treated by a team of five microsurgeons, were enrolled (Figure 1).

A modified Clavien–Dindo classification that included the new grade “IIIc,” for “’partial or total free flap failure after intervention’ needing further surgery in general anesthesia,” was applied to rate postoperative complications. Events leading to unexpected second surgery were also documented, including hematoma evacuation (donor or recipient site), wound dehiscence (donor or recipient site), vascular complication (donor or recipient site), lymph leakage (donor or recipient site), and fistula formation (Table 1).

The following patient characteristics, preexisting comorbidities, and cancer statuses were recorded for outcome prediction: age; sex; body mass index; smoking; alcoholism; diabetes mellitus; hypertension; chronic kidney disease (CKD, grade III–V); history of operation, radiotherapy, or chemotherapy for head and neck malignancy; T-stage; N-stage; overall cancer stage; tracheostomy; neck dissection; operation time; blood loss; American Society of Anesthesiologists classification; and Charlson Comorbidity Index. The Charlson Comorbidity Index was first proposed in 1987 to predict the 10-year survival in patients with multiple comorbidities [14,15], including age, myocardial infarction, congestive heart failure, peripheral vascular disease, cerebrovascular accident, dementia, chronic obstructive pulmonary disease, connective tissue disease, peptic ulcer disease, liver disease, diabetes mellitus, hemiplegia, CKD, localized solid tumor, leukemia or lymphoma, and acquired immune deficiency syndrome. Clinical outcomes, including the duration of hospital stay, duration of ICU stay, number of reoperations, and wound infection rate, for different severity levels (grade IIIa, grade IIIb, grade IIIc, and grade IVa) were investigated.

### Statistical Analysis

The demographic and clinical characteristics of patients in grade IIIb and IIIc were compared using the chi-square test for nominal variables (number and percentage) and the independent sample t-test for continuous variables (mean and standard deviation). Factors associated with the severity level with a *p* value of less than 0.05 were further investigated using logistic regression for categorical variables and linear regression for continuous variables. The continuous variables trends from grade IIIa to grade IVa, which were associated with the severity levels, were analyzed using linear contrast in the general linear model. The Cochran–Armitage trend test was used for categorical variables. *p* < 0.05 was considered statistically significant.

## 3. Results

The patients were grouped into grade IIIa (4 patients, 3.4%), IIIb (77 patients, 66.4%), IIIc (32 patients, 27.6%), and IVa (3 patients, 2.6%) according to the modified Clavien–Dindo classification system. Events causing unplanned operations were hematoma evacuation (34 patients, 29.3%), wound dehiscence (24 patients, 20.7%), vascular complications (93 patients, 80.2%), lymph leakage (2 patients, 1.7%), and fistula formation (6 patients, 5.2%) (Table 1).

Perioperative factors for outcome prediction were analyzed, and the incidence of alcoholism was higher in the grade IIIb group (72.7%) than in the grade IIIc group (50%; *p* = 0.028). Age, sex, body mass index, preexisting comorbidities, surgical etiology, cancer status, and associated surgical procedures did not differ significantly between the grade IIIb and grade IIIc groups. The Charlson Comorbidity Index (*p* = 0.219) also did not differ significantly between the two groups (Table 2).

A significant difference was observed between the grade IIIb and grade IIIc groups in several measures of complication severity, including the duration of hospital stay (23.1 ± 10.1 days in grade IIIb vs. 28.6 ± 11.9 days in grade IIIc, *p* = 0.015), duration of ICU stay (6.0 ± 3.4 days in grade IIIb vs. 8.7 ± 4.3 days in grade IIIc, *p* = 0.001), number of reoperations when hospitalized (1.4 ± 0.8 times in grade IIIb vs. 2.0 ± 1.0 times in grade IIIc, *p* < 0.001), and wound infection rate (29.9% in grade IIIb vs. 62.5% in grade IIIc, *p* = 0.002). The wound infection rate was also strongly associated with an increased severity from grade IIIb to grade IIIc (odds ratio = 3.91; 95% confidence interval = 1.65–9.30; *p* = 0.002). Furthermore, the number of reoperations during the current hospitalization (regression coefficient, 2.21 ± 0.76; *p* < 0.001), the duration of ICU stay (regression coefficient, 2.71 ± 1.54; *p* = 0.001), and the duration of hospital stay (regression coefficient, 0.2 ± 0.05; *p* < 0.001) were significantly positively correlated with grades IIIb and IIIc (Table 3).

A rising trend was observed in the duration of ICU stay (*p* < 0.001), duration of hospital stay (*p* = 0.001), and wound infection rate (*p* = 0.002) from grade IIIa to grade IVa, which represents a correlation between the severity grading levels of postoperative complications and patient outcomes (Figure 2A–C). The number of reoperations during the current hospitalization within the newly modified classification also increased in the Box–Whisker plot picture (Figure 2D).

## 4. Discussion

Various factors are reportedly associated with the development of postoperative complications and the need for further operation (Clavien–Dindo classification grade III), including age, smoking status, alcohol consumption, American Society of Anesthesiologists classification, T-stage, tumor location, free flap type, type of neck dissection, duration of operation, quantity of blood loss, advanced lymph node invasion (≥N_2_), and previous treatment with primary radiotherapy or chemoradiation for head and neck malignancy [3,10,11,12,16,17,18,19,20,21]. Broome et al. also indicated that low-volume surgeons and rarely-used free flaps were prone to develop severe complications (Clavien–Dindo classification grade III or above) and prolonged hospitalization [18]. McMahon et al. discovered that preoperative activated systemic inflammatory response syndrome, measured by elevated C-reactive protein and decreased serum albumin level, is associated with postoperative complications [22]. Thus, adequate perioperative management is crucial to the preoperative optimization of patients’ status. Moreover, the magnitude of the surgery, measured by prolonged operation time, reconstruction with free flap containing bone, and blood transfusion, also predicted postoperative complications [11,19,21,22].

The free flap outcome (Modified Clavien–Dindo classification grade IIIb and grade IIIc) is critical for success in head and neck cancer reconstruction. However, several factors may affect the prognosis, which causes identifying predictive factors for the free flap outcomes to be difficult. In our present study, most perioperative factors are not prognostic of the free flap outcome after an unplanned operation. While the Charlson Comorbidity Index is not predictive of the free flap outcome in our study, Perisanidis et al. reported that Charlson’s comorbidity index has an impact on total flap loss lead by vascular compromise [3]. Shayan et al. determined that preoperative irradiation with more than 60 Gy significantly increased the risk of flap loss, but the correlation of preoperative radiotherapy with free flap failure remains contentious [23,24,25]. We also did not observe a significant correlation.

Patients with malnutrition and sarcopenia are prone to postoperative complications, especially infections and healing problems [26,27]. Among the factors investigated in the present study, only alcoholism was predictive of the flap being saved (grade IIIb) or of partial or total failure (grade IIIc). Previous studies have demonstrated that the serum prealbumin level was a reliable marker for early alcoholic cirrhosis [28,29]. Moreover, Shum et al. determined that low prealbumin levels were associated with microvascular free flap failure [30]. Perhaps the better outcome in alcoholics could result from the higher prealbumin levels in alcoholics. We hypothesized that alcoholism caused restlessness and irritation in patients, which can cause vessel compression or twisting because of neck movement. Vessel compression and twisting could be reexplored before vessel occlusion under effective flap surveillance. The flap can usually be salvaged with early reoperation [31,32]. Additionally, patients with alcoholism would have reduced liver functionality, which may lead to low platelets level and prolonged coagulation time. It could help to prevent thrombus formation, a major cause of free flap failure, and, therefore, a more salvageable condition (grade IIIb). However, HK Kao et al. displayed that there was no correlation between the patency of microvascular anastomosis and the severity of liver cirrhosis, but a higher risk of hematoma formation in advanced cirrhosis [33,34]. The correlation between free flap loss and markers of liver function, such as aspartate transaminase, alanine transaminase, total bilirubin, direct bilirubin, prolongation of prothrombin time, and activated partial thromboplastin time (aPTT), as well as albumin levels, could be analyzed in future studies.

Perisanidis et al. first adopted the Clavien-Dindo classification to grade and describe postoperative complications in head and neck cancer patients receiving reconstruction with a jejunal free flap in 2011 [3]. In 2014, a prospective cohort study performed by McMahon et al. aimed to understand the characteristics of postoperative complications after head and neck surgery with free flap repair and to evaluate the use of Clavien–Dindo classification [22]. Both studies seek to analyze the predictors and outcomes of postoperative complications, and they found the Clavien-Dindo classification suitable for head and neck free flap surgery. The free flap prognosis is critical in head and neck microsurgery, but there was no assessment concerning the effect of free flap loss on outcomes in both studies.

The modification of the Clavien–Dindo classification was first described by Broome et al. in 2016 to assess the perioperative factors associated with severe complications of head and neck free flap transfer [18]. However, it was only used to discriminate the minor and major complications of different clinical situations. Without explicitly testing the correlation between severity level and outcome, the application of this classification is limited. In 2019, Ebner et al. reported commonly seen postoperative minor wound complications in head and neck free flap transfer surgery [21]. These minor complications could be over-graded to grade IIIa by the original Clavien–Dindo classification. Therefore, they redefined the grade I and grade II based on the specific treatments, including simple wound rinsing and cleaning, seroma drainage, single suture, and superficial abscess drainage, etc. The treatment of minor wound complications is usually performed in a regular examination room or ward under local or no anesthesia and has little or no effect on patients’ health status, hospitalization, recovery, and health care costs, which causes this modified grading system to be less valuable for application. The Comprehensive Complications Index^®^ (CCI^®^) was another tool based on the Clavien–Dindo classification to summarize overall postoperative morbidity, and Ebner et al. first applied the CCI^®^ to evaluate head and neck microsurgery and revealed its reliability [21].

Postoperative medical complications of free flap reconstruction in patients with head and neck cancer significantly diminish the quality of life of patients and increase the risk of another free flap [35]. Additional operations usually engender prolonged hospital stays, high costs, delayed adjuvant treatments, and less favorable oncological prognoses [12,19]. Therefore, precise, objective, and broadly accepted grading systems for operative complications are crucial for recognizing preoperative risk factors and predicting clinical sequela. Moreover, a grading system would be valuable for evaluating surgical outcomes and providing a consistent interpretation or clinical pathways between medical practitioners and centers [1,2,36]. Because the survival of the flap is crucial in this operation, we included a new subgrade “IIIc” under the framework of the Clavien–Dindo classification, which distinguishes partial or total flap loss, for a more precise assessment of the severity level and the related outcomes (Table 4). Our findings confirmed a significant increase in the number of reoperations during the current hospitalization, percentage of wound infection, and duration of ICU and hospital stay in the grade IIIc group, which would increase the hospital costs. The different severity levels of grade IIIa, IIIb, IIIc, and IVa accorded with the eventual surgical outcomes. This new classification is easily practicable, accounts more accurately for clinical significance, and provides a convincing tool for quality assessment in head and neck surgery.

This study has several limitations. First, this is a retrospective study, and the available data may be biased. Second, factors associated with the severity level can only indirectly reflect the real cost. The correlation between the total cost of hospitalization and the severity level requires statistical confirmation. Third, both grade IIIa (four patients) and IVa (three patients) involve only a small sample size, which may affect the reliability of the study. Fourth, the comparison of data between the modified and the standard Clavien-Dindo classification requires further investigation. The investigation of this modified complication-grading system in several global medical centers is still essential for validation.

## 5. Conclusions

The free flap prognosis is critical in head and neck surgery outcomes. Most preexisting comorbidities, characteristics, and preoperative cancer status of patients with head and neck cancer did not accurately predict the prognosis of the free flap after second surgery. The modified Clavien–Dindo classification system with grade IIIc for “partial or total free flap failure after intervention” needing further surgery under general anesthesia reflects different severity levels that are correlated with eventual clinical outcomes, including the duration of hospital stay, duration of ICU stay, wound infection rate, and number of reoperations.

## Figures and Tables

**Figure 1 jcm-09-03770-f001:**
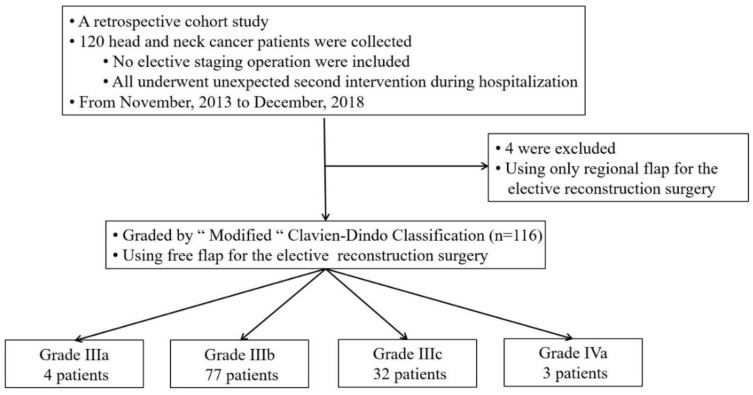
Flowchart of patient enrollment.

**Figure 2 jcm-09-03770-f002:**
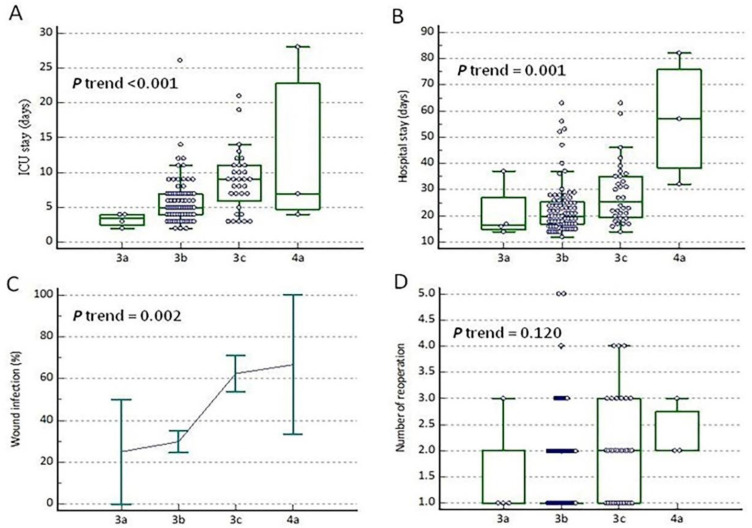
The Box-Whisker plot of factors associated with the severity level from grade IIIa to grade IVa, including the duration of intensive care unit stay (**A**), duration of hospital stay (**B**), percentage of wound infection (**C**), and the number of reoperations during the current hospitalization (**D**). All four factors exhibited a rising trend from grade IIIb to grade IIIc, which represented an increasing severity level of post-operative complications. The error bar represents the standard error.

**Table 1 jcm-09-03770-t001:** Complications requiring unplanned second surgery.

Complications	Grade IIIa	Grade IIIb	Grade IIIc	Grade IVa	Total
Hematoma evacuation	0	29 (37.7%) *	5 (15.6%)	0	34 (29.3%)
Wound dehiscence	3 (75%) *	12 (15.6%)	8 (25%)	1 (33.3%)	24 (20.7%)
Vascular complication	1 (25%)	57 (74%) *	32 (100%)	3 (100%)	93 (80.2%)
Lymph leakage	0	2 (2.6%) *	0	0	2 (1.7%)
Fistula formation	0	3 (3.9%)	2 (6.3%)	1 (33.3%)	6 (5.2%)

* Including donor site complications.

**Table 2 jcm-09-03770-t002:** Demographic and clinical characteristics of patients with grade IIIb and grade IIIc.

	Grade IIIb	Grade IIIc	*p*
Characteristics			
Patient number	77	32	-
Age (years)	56.3 ± 10.6	56.0 ± 11.1	0.901
Male	72 (93.5)	27 (84.4)	0.154
Body mass index (kg/m^2^)	23.7 ± 4.6	24.8 ± 3.3	0.263
Current or former smoker	62 (80.5)	24 (75.0)	0.608
Alcoholism	56 (72.7)	16 (50.0)	0.028 *
Comorbidity			
Diabetes mellitus	23 (29.9)	10 (31.3)	1.000
Hypertension	37 (48.1)	14 (43.8)	0.833
ASA classification			1.000
1–2	40 (51.9)	17 (53.1)	
3–4	37 (48.1)	15 (46.9)	
Charlson Comorbidity Index score	4.2 ± 1.7	3.8 ± 1.5	0.219
Previous operation	29 (37.7)	19 (59.4)	0.056
Previous C/T	22 (28.6)	11 (34.4)	0.648
Previous R/T	26 (33.8)	12 (37.5)	0.826
Surgical etiology			
Primary malignancy	49 (63.6)	14 (43.8)	0.088
Recurrent/Residual cancer	20 (26.0)	14 (43.8)	0.075
Previous free flap reconstruction	10 (13.0)	5 (15.6)	0.763
Cancer status			
T stage			0.520
1–2	32 (42.7)	11 (34.4)	
3–4	43 (57.3)	21 (65.6)	
N stage			0.667
0–1	40 (55.6)	19 (61.3)	
>1	32 (44.4)	12 (38.7)	
Overall cancer stage			0.321
Early	19 (25.3)	5 (15.6)	
Advanced	56 (74.7)	27 (84.4)	
Associated surgical procedures			
Tracheostomy	64 (83.1)	27 (84.4)	1.000
Neck dissection	55 (71.4)	18 (56.3)	0.179

ASA, The American Society of Anesthesiologists; C/T, chemotherapy; R/T, radiotherapy; OP, operation; ICU, intensive care unit. The pathological cancer stage was based on the AJCC 2018 guideline. Nominal variables are expressed in numbers and percentages and were compared using the chi-square test. Continuous variables are presented as means ± SDs and were compared using the independent sample t-test. * *p* < 0.05.

**Table 3 jcm-09-03770-t003:** Outcome comparison between grade IIIc and grade IIIb.

Outcome	Grade IIIb	Grade IIIc	*p*	*B* or Odds Ratio (95% CI)	*p*
Categorical variable					
Wound infection (%)	23 (29.9)	20 (62.5)	0.002 *	3.91 (1.65–9.30)	0.002 *
Continuous variable					
Reoperation times during the current hospitalization	1.4 ± 0.8	2.0 ± 1.0	<0.001 *	2.21 (1.45, 2.97)	<0.001 *
ICU stay (day)	6.0 ± 3.4	8.7 ± 4.3	0.001 *	2.71 (1.17, 4.24)	0.001 *
Hospital stay (day)	23.1 ± 10.1	28.6 ± 11.9	0.015 *	0.20 (0.15, 0.25)	<0.001 *

*B*, regression coefficient; CI, confidence interval; ICU, intensive care unit. Categorical variables are expressed as numbers and percentages and were compared using the chi-square test and logistic regression. Continuous variables are presented as means ± SDs and were compared using the independent sample t-test and linear regressions. * *p* < 0.05.

**Table 4 jcm-09-03770-t004:** Modified Clavien–Dindo classification for free flap reconstruction in patients with head and neck cancer.

Grade	Definition
I	Any deviation from the normal postoperative course without the need for pharmacological treatment or surgical, endoscopic, or radiological interventions
II	Requiring pharmacological treatment with drugs other than those permitted for grade I complication (would include blood transfusions and total parenteral nutrition)
III	Requiring surgical, endoscopic, or radiological intervention
IIIa	Intervention not under general anesthesia
IIIb	Intervention under general anesthesia
IIIc	“Partial or total free flap failure after intervention” needing further surgery under general anesthesia
IV	Life-threatening complication (including complications of the central nervous system) that requires management in a high dependency or intensive therapy unit
IVa	Single organ dysfunction (including dialysis)
IVb	Multiorgan dysfunction
V	Death

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
