# Peer review of "Modified Clavien–Dindo Classification and Outcome Prediction in Free Flap Reconstruction among Patients with Head and Neck Cancer"

_jcm, 2020, doi:10.3390/jcm9113770_

Round 1
Reviewer 1 Report
Dear authors,
General Comments
Overall, this retrospective cohort study deals with the outcome prediction in free flap reconstruction among patients with Head and Neck cancer using an modified Clavian-Dindo classification and is very well designed and presented. The intention to improve the existing the Clavian-Dindo classification is of great clinical interest.
Minor comments
- The initial description of Clavien-Dindo classification is missing and should be added as comparison in the introduction.
- The number of patients assigned to the modified Clavien-Dindo classification Grade IIIa and IVa is very small. This should be mentioned in the limitations.
Author Response
Point 1: The initial description of Clavien-Dindo classification is missing and should be added as comparison in the introduction.
Response 1:
Thanks for your suggestion. The initial description of Clavien-Dindo classification has been added in the introduction (Row 33-44).
“It was first described by Clavien PA and Dindo D in 2004 [1]. Grade I of the Clavien–Dindo classification was defined as any deviation from the normal postoperative course without the need for pharmacological treatment or surgical, endoscopic, or radiological interventions. Acceptable regimens include antiemetics, analgesics, antipyretics, diuretics, and bedside wound care. Grade II was defined as complications requiring pharmacological treatment with drugs other than those permitted for grade I complication. Grade III of the Clavien–Dindo classification was originally defined as surgical complications requiring surgical, endoscopic, or radiological intervention and was further classified into grade IIIa and IIIb depending on whether the intervention was performed under general anesthesia. Grade IV was defined as a life-threatening complication that requires management in a high dependency and was further classified into grade IVa and IVb in terms of single or multi-organ dysfunction. Finally, a complication resulting in death was graded as V.”
Point 2: The number of patients assigned to the modified Clavien-Dindo classification Grade IIIa and IVa is very small. This should be mentioned in the limitations.
Response 2:
Thanks for your kindly remind. We have added the description in the limitations (Row 196-197).
“Third, both grade IIIa (4 patients) and IVa (3 patients) involve only a small sample size, which may affect the reliability of the study.”

Reviewer 2 Report
This article states that the Clavien-Dindo classification is not suitable for head and neck surgery and has therefore to be modified. That is why, the authors added an additional grade „IIIc“ for „partial or total free flap failure“. This should help to discriminate relatively minor complications like hematoma evacuation or wound dehiscence from severe complications like partial or total free flap loss resulting in an additional salvage surgery or anew free flap surgery, that would have been categorized in the the same grade (IIIb). To show that this discrimination is useful, they investigated if the outcome (regarding e.g. length of ICU/hospital stay) differs significantly between grade IIIb and IIIc.
Therefore, they used this modified classification to classify 116 patients who had undergone free flap reconstruction due to head and neck cancer. This population was selected retrospectively and criterion was an unexpected second surgery during hospitalisation after (successful) free flap reconstruction, so that all 116 were classified in grade III or higher. In the following, they mainly focused on the difference between grade IIIb and IIIc regarding complications leading to a second surgery like hematoma, wound dehiscence, vascular complication, lymph leakage and consequences during the course of hospitalisation like ICU stay, wound infection, reoperation times and complete hospital stay. Additionally, they searched for predictive factors leading to classification in grade IIIb or IIIc like age, sex, comorbidities…
They used the chi-square test for nominal variables and t-test for continous variables. Significant (p<0.05) factors were then investigated using logistic regression or linear regression for categorical variables or continuous variables, respectively. The continuous variables, which were associated with the severity level of complications were then analyzed using linear contrast in the general linear model and categorical variables were analyzed using the Cochran-Armitage test.
Regarding predictive factors leading to classification in grade IIIb or IIIc, they found that only alcoholism showed a significant effect. Interestingly, alcoholism seems to lead to minor complications as the incidence of alcoholism was 72.7% in grade IIIb patients and 52% in grade IIIc patients. They hypothesized that this could be due to a restlessness and irritation in patients, which then leads to vessel compression or twisting because of the neck, movement. Under sufficient flap surveillance this would lead to earlier re-exploration and therefore to successful salvage surgery. This is difficult to understand. This would mean, that reversible factors, like compression of the anastomosis due to neck movement would lead to a unnecessary „successful“ revision surgery. Maybe, patients with alcoholism have a reduced liver functionality and therefore maybe they have low platelets and a reduced coagulation which could help to prevent thrombosis, the most often factor for free flap failure (all 32 patients in this study had vascular complication).
Regarding the usefulness of a modified Clavien-Dindo classification they found that complications following the second surgery differ significantly between grade IIIb and IIIc. This justifies the modification and implementation of an additional grade for partial or total free flap loss. Nevertheless, it stays unclear where there is the difference between this modification and the one by Broome et al. in 2016.
Broome et al. added one additional category “distinguishing between patients requiring a surgical, endoscopic or radiological intervention for hematoma evacuation without wound-healing complications, minor wound dehiscence, or vascula complication with complete functional salvage of the flap and those requiring intervention for fistulas, or complete or partial flap loss resulting in additional procedures” .(Broome et al. 2016)
In this paper, they put in a grade “IIIc” for “partial or total free flap failure” and could affirm the results of Broome et al. that a differentiation is necessary.
The article is written in comprehensible English, there are just a few words that need a second look.
Row 40 „unique“: This suggests that these complications (hematoma, flap loss, fistula) only exist in head and neck free flap surgery. It would be better to use „specific“.
Row 59/68/figure 1 „secondary“: Secondary surgery means that there is a primary surgery that should be performed first. In this context I suppose they mean „second“.
A definition of partial or free flap loss would clarify some uncertainties. How is partial loss defined? When is total/partial free flap loss determined? On the date of second surgery? Is a free flap loss occurring after a possible second surgery due to .e.g. hematoma evacuation counted to group IIIc? What period is taken into account?
To sum up, this paper shows that adding the new grade “partial or total free flap loss” in the Clavien-Dindo classification seems to be useful in head and neck microvascular surgery. The authors show that complications like duration of ICU/hospital stay, wound infection rate and numbers of reoperations are correlated with the grade in their modified Clavien-Dindo classification. They could not find a predictive factor. The modification seems to be similar to the one Broome et al. did in 2016.
Author Response
Point 1: Regarding predictive factors leading to classification in grade IIIb or IIIc, they found that only alcoholism showed a significant effect. Interestingly, alcoholism seems to lead to minor complications as the incidence of alcoholism was 72.7% in grade IIIb patients and 52% in grade IIIc patients. They hypothesized that this could be due to a restlessness and irritation in patients, which then leads to vessel compression or twisting because of the neck, movement. Under sufficient flap surveillance this would lead to earlier re-exploration and therefore to successful salvage surgery. This is difficult to understand. This would mean, that reversible factors, like compression of the anastomosis due to neck movement would lead to an unnecessary “successful” revision surgery. Maybe, patients with alcoholism have a reduced liver functionality and therefore maybe they have low platelets and a reduced coagulation which could help to prevent thrombosis, the most often factor for free flap failure (all 32 patients in this study had vascular complication).
Response 1:
Thank you for this comment. Indeed, our initial hypothesis would indicate that alcoholism indirectly results in an unnecessary second operation. We agree with your advice, and the content has been revised (Row 152-160) with new reference #29 and #30.
“We hypothesized that alcoholism caused restlessness and irritation in patients, which can cause vessel compression or twisting because of neck movement. Vessel compression and twisting could be reexplored before vessel occlusion under effective flap surveillance. The flap can usually be salvaged with early reoperation [27,28]. Additionally, patients with alcoholism would have reduced liver functionality, which may lead to low platelets level and prolonged coagulation time. It could help to prevent thrombus formation, a major cause of free flap failure, and, therefore, a more salvageable condition (grade IIIb). However, HK Kao et al. displayed that there was no correlation between the patency of microvascular anastomosis and the severity of liver cirrhosis, but a higher risk of hematoma formation in advanced cirrhosis [29,30].”
Point 2: Regarding the usefulness of a modified Clavien-Dindo classification they found that complications following the second surgery differ significantly between grade IIIb and IIIc. This justifies the modification and implementation of an additional grade for partial or total free flap loss. Nevertheless, it stays unclear where there is the difference between this modification and the one by Broome et al. in 2016.
Broome et al. added one additional category “distinguishing between patients requiring a surgical, endoscopic or radiological intervention for hematoma evacuation without wound-healing complications, minor wound dehiscence, or vascula complication with complete functional salvage of the flap and those requiring intervention for fistulas, or complete or partial flap loss resulting in additional procedures”. (Broome et al. 2016)
In this paper, they put in a grade “IIIc” for “partial or total free flap failure” and could affirm the results of Broome et al. that a differentiation is necessary.
Response 2:
The free flap prognosis is critical in head and neck surgery outcomes. We aimed to modify the Clavien-Dindo classification under the original framework; by adding grade “IIIc” for “partial or total free flap failure requiring intervention.” Broome et al. aimed to identify a correlation of several predictive factors with severe complications in head and neck free flap reconstruction. The classification proposed by Broome et al. was only used to discriminate the minor and major complications of different clinical situations. However, there was insufficient statistical evidence for a correlation between the severity level of the classification and the prognosis in their study, which limited the application.
Point 3: The article is written in comprehensible English, there are just a few words that need a second look.
Row 40 „unique“: This suggests that these complications (hematoma, flap loss, fistula) only exist in head and neck free flap surgery. It would be better to use „specific“.
Row 59/68/figure 1 „secondary“: Secondary surgery means that there is a primary surgery that should be performed first. In this context I suppose they mean „second“.
Response 3:
Thanks for your kindly remind. We have revised the words according to your advice, and it reveals a more accurate meaning (Row 48/67/77/ Figure 1).
Point 4: A definition of partial or free flap loss would clarify some uncertainties. How is partial loss defined? When is total/partial free flap loss determined? On the date of second surgery?
Response 4: If any part of the flap is preserved, it is defined as partial flap loss, which requires intervention such as debridement and/or flap (local/pedicle/free) reconstruction. Total/partial free flap loss is determined on the date of the second surgery according to the free flap status at that time.
Point 5: Is a free flap loss occurring after a possible second surgery due to .e.g. hematoma evacuation counted to group IIIc? What period is taken into account?
Response 5: It should count to grade IIIc if a free flap loss occurred, no matter what the cause was.

Reviewer 3 Report
improving classification for analysis of Head and Neck surgical outcomes is of great importance and this paper should encourage discussion
Author Response
Point 1: Improving classification for analysis of Head and Neck surgical outcomes is of great importance and this paper should encourage discussion.
Response 1:
We appreciate your recognition and encouragement. The free flap prognosis is critical in head and neck surgery outcomes. The modified Clavien–Dindo classification system with grade IIIc for partial or total free flap failure reflects different severity levels that are correlated with eventual clinical outcomes.

Round 2
Reviewer 2 Report
Overall, the article is written in comprehensible English and presents the result of this retrospective evaluation adequately.
The introduction provides a good start and the problem is stated clearly. They explain their methods understandably and present the results in a sufficient way.
The discussion is rather weak. It would have been nice to at least mention the work of McMahon et al. in 2014 (https://www.sciencedirect.com/science/article/pii/S0266435613001563?via%3Dihub) who did a prospective cohort study regarding postoperative complications after head and neck free flap surgery using the Clavien-Dindo-classification as well as to discuss the work of Perisanidis et al. in 2011 ( https://linkinghub.elsevier.com/retrieve/pii/S0266435611000337 ) who used a modified Clavien-Dindo classification in head and neck microsurgery patients receiving a jejunal flap. Both found the Clavien-Dindo classification suitable for head and neck free flap surgery.
In addition, it would be nice if a comparison of the data between the modified and the standard Clavien-Dindo classification would have been made as Ebner et al. in 2019 suggested. Additionally, the Comprehensive Complication Index (CCI®) would have been worth being mentioned.
A differentiation between partial and total flap loss could be useful, since not every partial flap loss needs a further/third surgery in general anaesthesia, which is probably the reason for longer hospital stay etc. Alternatively, maybe the definition of grade IIIc could be modified: “partial or total free flap failure after intervention” needing further surgery in general anaesthesia? For the reader it stays unclear, whether partial free flap loss (e.g. superficial necrosis with following bedside necrosectomy) is classified in grade IIIc or IIIa, for example. Not every (partial) flap loss needs another surgery.
Then, it stays unclear where lies the difference to the modification of Broome et al. and this work. Of course, Broome et al. aimed for factors influencing the incidence of severe complication in head and neck free flap reconstructions, still they introduced this modification first. At least, it should be mentioned in the discussion that the modification was first described by Broome et al. without explicitly testing the correlation between severity level and outcome. This has been done in this work.
It would be nice to discuss the differences in predictors compared to other studies (Broome, McMahon…).
Regarding the prealbumin levels, maybe another explanation for the better outcome in alcoholics could be the higher prealbumin levels in alcoholics ? (https://www.sciencedirect.com/science/article/pii/000989818390030X ;
https://www.sciencedirect.com/science/article/pii/0741832985901041 )
All in all the discussion lacks discussion with the available literature, and interesting literature is even not mentioned.
Table 1: secondary --> second
line 203: secondary --> second
Author Response
Point 1: It would have been nice to at least mention the work of McMahon et al. in 2014 who did a prospective cohort study regarding postoperative complications after head and neck free flap surgery using the Clavien-Dindo-classification as well as to discuss the work of Perisanidis et al. in 2011 who used a modified Clavien-Dindo classification in head and neck microsurgery patients receiving a jejunal flap. Both found the Clavien-Dindo classification suitable for head and neck free flap surgery.
Response 1:
Thank you for the suggestion. We agree with your advice, and the content has been revised (Row 183-191) with new reference # 22.
“Perisanidis et al. first adopted the Clavien-Dindo classification to grade and describe postoperative complications in head and neck cancer patients receiving reconstruction with a jejunal free flap in 2011 [3]. In 2014, a prospective cohort study performed by McMahon et al. aimed to understand the characteristics of postoperative complications after head and neck surgery with free flap repair and to evaluate the use of Clavien–Dindo classification [22]. Both studies seek to analyze the predictors and outcomes of postoperative complications, and they found the Clavien-Dindo classification suitable for head and neck free flap surgery. The free flap prognosis is critical in head and neck microsurgery, but there was no assessment concerning the effect of free flap loss on outcomes in both studies.”
Point 2: In addition, it would be nice if a comparison of the data between the modified and the standard Clavien-Dindo classification would have been made as Ebner et al. in 2019 suggested. Additionally, the Comprehensive Complication Index (CCI®) would have been worth being mentioned.
Response 2:
Thank you for the suggestion. However, there was currently no data regarding the comparison between the modified and standard Clavien-Dindo classification in our study, and this would be added to the limitations (Row 227-228). The application of the Comprehensive Complication Index (CCI®) has been mentioned in the content (Row 204-206).
Row 227-228
“Fourth, the comparison of data between the modified and the standard Clavien-Dindo classification requires further investigation.”
Row 204-206
“The Comprehensive Complications Index® (CCI®) was another tool based on the Clavien–Dindo classification to summarize overall postoperative morbidity, and Ebner et al. first applied the CCI® to evaluate head and neck microsurgery and revealed its reliability [21].”
Point 3: A differentiation between partial and total flap loss could be useful, since not every partial flap loss needs a further/third surgery in general anaesthesia, which is probably the reason for longer hospital stay etc. Alternatively, maybe the definition of grade IIIc could be modified: “partial or total free flap failure after intervention” needing further surgery in general anaesthesia? For the reader it stays unclear, whether partial free flap loss (e.g. superficial necrosis with following bedside necrosectomy) is classified in grade IIIc or IIIa, for example. Not every (partial) flap loss needs another surgery.
Response 3:
Thanks for your advice. The definition of grade IIIc is modified as” ’partial or total free flap failure after intervention’ needing further surgery in general anaesthesia.” It is clearer for the reader to classify partial free flap loss based on whether the surgery is performed under general anaesthesia.
Point 4: Then, it stays unclear where lies the difference to the modification of Broome et al. and this work. Of course, Broome et al. aimed for factors influencing the incidence of severe complication in head and neck free flap reconstructions, still they introduced this modification first. At least, it should be mentioned in the discussion that the modification was first described by Broome et al. without explicitly testing the correlation between severity level and outcome. This has been done in this work.
Response 4:
It has been mentioned in the content (Row 192-196).
“The modification of the Clavien–Dindo classification was first described by Broome et al. in 2016 to assess the perioperative factors associated with severe complications of head and neck free flap transfer [18]. However, it was only used to discriminate the minor and major complications of different clinical situations. Without explicitly testing the correlation between severity level and outcome, the application of this classification is limited.”
Point 5: It would be nice to discuss the differences in predictors compared to other studies (Broome, McMahon…).
Response 5:
Thanks for your suggestion. The content regarding the discussion of the differences in predictors has been added (Row 145-153 & Row 158-160) with new reference # 22.
Row 145-153:
“Broome et al. also indicated that low-volume surgeons and rarely-used free flaps were prone to develop severe complications (Clavien–Dindo classification grade III or above) and prolonged hospitalization [18]. McMahon et al. discovered that preoperative activated systemic inflammatory response syndrome, measured by elevated C-reactive protein and decreased serum albumin level, is associated with postoperative complications [22]. Thus, adequate perioperative management is crucial to the preoperative optimization of patients' status. Moreover, the magnitude of the surgery, measured by prolonged operation time, reconstruction with free flap containing bone, and blood transfusion, also predicted postoperative complications [11,19,21,22].”
Row 158-160:
“While the Charlson Comorbidity Index is not predictive of the free flap outcome in our study, Perisanidis et al. reported that Charlson’s comorbidity index has an impact on total flap loss lead by vascular compromise [3].”
Point 6: Regarding the prealbumin levels, maybe another explanation for the better outcome in alcoholics could be the higher prealbumin levels in alcoholics ? (https://www.sciencedirect.com/science/article/pii/000989818390030X ;
https://www.sciencedirect.com/science/article/pii/0741832985901041 )
Response 6:
Thank you for this comment. Indeed, the better outcome in alcoholics may result from the higher prealbumin levels in alcoholics. The content has been revised (Row 167-170) with new reference #28 and #29.
“Previous studies have demonstrated that the serum prealbumin level was a reliable marker for early alcoholic cirrhosis [28,29]. Moreover, Shum et al. determined that low prealbumin levels were associated with microvascular free flap failure [30]. Perhaps the better outcome in alcoholics could result from the higher prealbumin levels in alcoholics.”
Point 7: Table 1: secondary --> second
Line 203: secondary --> second
Response 7:
Thanks for your kindly remind. We have revised the words and it reveals a more accurate meaning (Table 1 / Row 234).
